# *Lactiplantibacillus plantarum* 1008 Enhances Testicular Function and Spermatogenesis via the Modulation of Gut Microbiota in Male Mice with High-Fat-Diet-Induced Obesity

**DOI:** 10.3390/biology13110890

**Published:** 2024-10-31

**Authors:** Chin-Yu Liu, Yi-Wen Chen, Tsung-Yu Tsai, Te-Hua Liu, Ting-Chia Chang, Chih-Wei Tsao

**Affiliations:** 1Department of Nutritional Science, Fu Jen Catholic University, New Taipei City 242062, Taiwan; nf351.lab@gmail.com (C.-Y.L.); angela55106577@gmail.com (Y.-W.C.); ctc5628@gmail.com (T.-C.C.); 2Department of Food Science, Fu Jen Catholic University, New Taipei City 242062, Taiwan; tytsai@mail.fju.edu.tw (T.-Y.T.); r19901217@hotmail.com (T.-H.L.); 3Division of Urology, Department of Surgery, Tri-Service General Hospital, National Defense Medical Center, Taipei 114202, Taiwan

**Keywords:** high-fat diet, probiotic, male infertility, oxidative stress, spermatogenesis

## Abstract

Many studies have noted that overweight and obese men are likely to suffer from infertility. Rodents fed high-fat diets develop obesity and infertility. Many of the unhealthy effects of obesity are thought to be linked to changes in the gut microbiota. The authors of this study decided to investigate whether feeding mice probiotics might alter their gut microbiota and in turn affect their infertility caused by a high-fat diet.

## 1. Introduction

Infertility, being defined as the inability to conceive following more than one year of regular unprotected sexual intercourse, affects 10–15% of couples during the reproductive age [1]. Male infertility is responsible for 20–30% of infertility cases and contributes to 50% of overall cases. Several studies have provided evidence on the causes of male infertility, from genetic mutations to lifestyle choices, medical illnesses, and medications. Furthermore, obesity is a common disease in modern society and an important factor causing male infertility. Studies have shown that overweight or obesity in males are associated with decreased sperm quality and a higher percentage of sperm with DNA fragmentation, therefore inducing male infertility [2,3]. Animal experiments have also demonstrated poor spermatogenesis [1], changes in hormone levels [4], and disrupted testicular function [5] caused by high-fat-diet (HFD)-induced obesity.

The mechanism through which obesity affects male reproduction is complex. Obesity is closely linked to male reproductive disorders through several mechanisms, including hormonal imbalances (such as lower testosterone levels), increased oxidative stress, and chronic inflammation, all of which negatively impact spermatogenesis and overall testicular function [6,7]. Previous studies have shown that oxidative stress, inflammation, and apoptosis are associated with impaired testicular function. Moreover, recent reports have indicated that the gut microbiota could influence spermatogenesis and metabolic function in HFD-induced obesity [8,9]. Ding et al. reported HFD-induced gut microbiota dysbiosis and defects in spermatogenesis with elevated endotoxin levels, while the transplantation of HFD-fed gut microbes into ND-maintained mice also decreased spermatogenesis and sperm quality; however, similar transplantation methods involving microbes from ND-fed mice failed to do so [10]. However, the role of the gut microbiota in the relationship between HFD-induced obesity and decreased spermatogenesis is still unclear.

The gut microbiome plays an important role in health maintenance and is closely related to metabolic function. *Lactiplantibacillus plantarum* 1008 (LP1008) is a Gram-positive bacterial probiotic. A probiotic is defined as “live microorganisms which when administered in adequate amounts confer a health benefit on the host”. Previous studies have concluded that the biological effects of LP1008 include attenuating HFD-induced abnormal metabolism, liver damage, and increased antioxidant ability, as well as reducing inflammation levels by regulating the gut microbiota [11,12]. Various interventions, including lifestyle modifications, pharmacological agents, and surgical procedures, have been explored to improve reproductive outcomes in obesity models [13]. However, these approaches have limitations; lifestyle changes require long-term adherence, and pharmacological treatments may produce adverse effects. While probiotic supplementation shows promise, further validation is necessary [14]. Due to fewer side effects and better patient adherence, nutraceuticals and dietary supplements have been investigated for their anti-obesity benefits. Despite this, limited studies focus on probiotics and the gut microbiota’s role in male reproduction. This study aims to evaluate the effects of LP1008 supplementation on reproductive damage induced by an HFD and the underlying mechanisms involved.

## 2. Materials and Methods

### 2.1. Animal Management and Experimental Groups

C57/BL6 male mice (10 weeks old, *n* = 36) were purchased from the National Laboratory Animal Center and housed in the laboratory animal center of the National Defense Medical Center. The animals were maintained on a 12 h light/dark cycle at a controlled temperature and relative humidity range of 23 ± 2 °C and 50–60%, respectively.

After a two-week acclimatization period to laboratory conditions, the mice were randomized into the following two groups: (a) a normal control group (NC group, *n* = 12), which received a normal diet (Laboratory Autoclavable Rodent Diet 5010) with 58% of the calories from carbohydrates, 29% from protein, and 13% from fat; and (b) a high-fat control group (HFC group, *n* = 24), which received a high-fat diet (Research Diets D12451) with 35% of the calories from carbohydrates, 20% from protein, and 45% from fat. After 16 weeks, the HFC group was randomly divided into the following two groups: a high-fat control (HFC) group and (c) a high-fat diet + LP1008 (HFP) group. The HFC group was maintained on the original diet, while the HFP group was fed the high-fat diet and administered a gavage of LP1008 (2.05 × 10^9^ CFU/kg BW/day), while the NC group and HFC group were given purified water via gavage. After 8 weeks of gavage, the mice were deeply anesthetized with a mixture of Zoletil 50 (containing 25 mg/mL tiletamine and zolazepam; Virbac, Carros, France) and Rompun (containing 23.32 mg/mL xylazine hydrochloride; Bayer, Taipei, Taiwan) administered intraperitoneally. Upon confirmation of adequate anesthesia through the absence of reflexes, exsanguination was conducted via cardiac puncture. A sterile 25-gauge needle attached to a 1 mL syringe was carefully inserted into the left ventricle, and blood was withdrawn slowly to prevent hemolysis. The collected blood was immediately processed for downstream analyses. This procedure ensured minimal stress and discomfort to the animals and was performed as part of humane euthanasia at the end of the experimental protocol.

### 2.2. Analyses of Biochemical Markers

After anesthetization, blood samples were collected and centrifuged at 2000× *g* for 20 min, and serum was isolated and stored at −80 °C until analysis. Total cholesterol (TC), triglyceride (TG), glucose, aspartate aminotransferase (AST), and alanine aminotransferase (ALT) levels were measured using a biochemistry automatic analyzer (Hitachi 7180, Tokyo, Japan). The levels of serum insulin (Mercodia Mouse Insulin ELISA, #10124701, Uppsala, Sweden), testosterone (MyBioSource, MBS#163127, San Diego, CA, USA), and LPS (Thermo Fisher, #A39552, Rockford, IL, USA) were measured using an ELISA kit according to the manufacturer’s instructions.

### 2.3. Analyses of Sperm Motility, Sperm Count, and Sperm Morphology

Sperm quality analysis included assessment of sperm motility, sperm count, and sperm morphological abnormalities. The vas deferens is dissected. Sperm is then collected by flushing or gently compressing the vas deferens. Sperm motility was assessed under a light microscope at 40× magnification (E400; Nikon, Tokyo, Japan) in 4 random fields, and at least 200 sperm cells were counted. The sperm count was measured in a diluted sperm suspension using an automated cell counter (TC20, Bio-Rad, Hercules, CA, USA). For sperm morphology analysis, sperm samples were incubated at 37 °C for 15 min, placed on a slide at room temperature, fixed with methanol (Honeywell, Morris Plains, NJ, USA), stained with an eosin Y (E4009, Sigma-Aldrich, Saint Louis, MO, USA) and ethanol (Bioman, Taipei City, Taiwan) mixture, rinsed with 75% ethanol (Bioman), and dried. The slides were evaluated under a light microscope (DM1000; Leica) to calculate the morphology percentages for at least 100 sperm cells.

### 2.4. Analyses of Histology

Formalin-fixed liver, testis, and kidney tissues were prepared and analyzed at the Department of Pathology in Cardinal Tien Hospital (New Taipei City, Taiwan) after tissue processing. The samples were observed and evaluated under a DM1000 light microscope (Leica, Wetzlar, Germany). The thickness of the germinal epithelium (GE), mean seminiferous tubule diameter (MSTD), and glomerular size were then measured using ImageJ software (1.50; National Institutes of Health, Bethesda, MD, USA). The GE thickness was calculated as the average length of a seminiferous tubule with the average length of a lumen subtracted. The degree of spermatogenesis was scored histologically using the Johnson score.

### 2.5. Analyses of Apoptosis-, Inflammation-, and Autophagy-Related Proteins in the Testes

The testes were stored at −80 °C after euthanasia. To prepare samples for Western blotting, testis tissues were lysed in a mixture of RIPA buffer, protease inhibitor, and phosphatase inhibitor (Thermo Fisher Scientific, Waltham, MA, USA), and the total protein was quantified using a DC protein assay (Bio-Rad). The extracted protein was loaded, separated by electrophoresis, and transferred onto polyvinylidene difluoride (PVDF) membranes. After being blocked in blocking buffer for 1 h, the membranes were cut, incubated with the primary antibody overnight, and washed 3 times in TBST buffer. Then, the membranes were incubated with an anti-mouse (1/4000; sc-2005, Santa Cruz Biotechnology, Paso Robles, CA, USA) or anti-rabbit secondary antibody (1/5000; sc-2005, Santa Cruz) for 1 h and detected using chemiluminescence. The primary antibodies used were against caspase-9 (1/1000, #9508; Cell Signaling Technology, Beverly, MA, USA), bax (1/1000, #2772; Cell Signaling), bcl-xl (1/1000, ab32370; Abcam, Waltham, MA, USA), caspase-3 (1/750, #9662; Cell Signaling), NF-ĸB (1/1000, E381; Abcam), TNF-α (1/1000, ab1793; Abcam), IL6 (1/500, sc-57315), Beclin 1 (1/1000, sc-48341), p62 (1/1000, sc-48402), LC3B-II (1/1000, sc-271625) (all from Santa Cruz Biotechnology), and β-actin (1/10,000, A5316; Sigma, St. Louis, MO, USA).

### 2.6. Analyses of the Redox Status in the Testis

Redox status, as demonstrated by the activities of antioxidants and the MDA content in the testes, was analyzed using assay kits purchased from Cayman (SOD, Item No. 706002, Ann Arbor, MI, USA; Catalase, Item No. 707002; GPx, Item No. 703102; TBARS, Item No. 10009055), following the protocols provided by the manufacturer.

### 2.7. 16S rRNA Sequencing with Illumina MiSeq Sequencing

For gut microbiota sampling, freshly frozen fecal samples were collected from mice after 8 weeks of LP1008 treatment. The fecal samples were stored at −80 °C to preserve microbial content. DNA was extracted from the feces using a standardized fecal DNA extraction kit. The 16S rRNA gene was amplified using specific primers for bacterial communities. The amplicon libraries were sequenced using the Illumina MiSeq platform (Genomics BioSci & Tech Co., New Taipei City, Taiwan). Paired-end reads (2 × 300 bp) were trimmed using Trimmomatics (v0.39) and demultiplexed using an in-house script. Sequences from both ends of the 341F-805R primers were trimmed using Cutadapt (v3.3) with the following criteria: read length ≥ 150 bp and an error rate of 0.1 as default. DADA2 (v1.12) was subsequently used to perform preprocessing, which included filtering out noisy sequences (denoise), merging paired-end reads, and removing chimeras to extract amplicon sequence variants (ASVs). The average demultiplexed sequence counts were 63,154, 60,400, and 58,685 in the NC, HFC, and HFP groups, respectively, and the average remaining sequence counts after denoising were 50,409, 49,113, and 47,979, respectively. The identified ASV sequences were annotated by a QIIME2 naive Bayes classifier (v2019.10), using the SILVA 132 99% identity as the reference. A Venn diagram (v1.6.17) was used to analyze both the common and unique information between different samples. The results of species annotation were interactively shown using KRONA (v2.7.1), as well as bar plots. The alpha diversity was analyzed using a species richness estimator (Chao1, observed species, Good’s coverage, and Fisher’s alpha) and a species evenness estimator (Shannon, Simpson, and ENSPIE). The beta diversity was analyzed using UniFrac distance matrices with a heatmap, hierarchical clustering, principal component analysis (PCA), and principal coordinate analysis (PCoA) with a Bray–Curtis similarity matrix.

### 2.8. Statistics

All the data are expressed as means and standard deviations and were analyzed using a one-way analysis of variance (ANOVA), followed by Duncan’s post hoc test with SAS Enterprise Guide 8.3 software. The results were considered significant at *p* < 0.05.

## 3. Results

### 3.1. LP1008 Ameliorates HFC-Induced Abnormal Serum Biochemical Levels

After 24 weeks of feeding, the HFC group presented higher serum concentrations of glucose, insulin, total cholesterol, and LPS than the NC group. *LP1008* treatment significantly decreased the serum insulin, total cholesterol, and LPS levels compared with those in the HFC group (Figure 1).

### 3.2. LP1008 Improved Testicular Spermatogenesis and Sperm Quality in the High-Fat Diet Group

Mice fed with HFD had significantly higher body weight before (NC: 32.1 ± 1.7; HFC: 44.4 ± 5.7; HFP: 44.5 ± 4.5) and after LP1008 treatment (NC: 32.2 ± 1.8; HFC: 48.0 ± 4.3; HFP: 47.5 ± 4.3). The results revealed significantly lower relative reproductive organ weights, including those of the testis, epididymis, and vas deferens, in the HFC group than in the NC group. In addition, the epididymal fat weight in the HFC group was significantly greater than in the NC group. However, LP1008 treatment resulted in no differences in reproductive organ weights between the HFC and HFP groups. After HFD exposure for 24 weeks, impaired spermatogenesis was clearly observed. The sperm quality parameters, such as count, motility, and normal morphology, significantly decreased in the HFC group, whereas LP1008 treatment significantly improved the sperm quality. Moreover, compared with those in the NC group, the MTBS and thickness of the GE were significantly greater in the HFC group, and LP1008 treatment significantly attenuated HFD-induced impaired spermatogenesis. With respect to the MSTD, there were no significant differences among the three groups (Figure 2).

### 3.3. LP1008 Increases Testicular Testosterone Levels by Up-Regulating 17β-HSD Expression

Compared with the NC group, the HFC group presented significantly greater decreases in serum and testicular testosterone concentrations and lower 17β-HSD levels. In contrast, the HFP group had significantly greater testicular testosterone levels; a similar increasing trend was also observed for the serum testosterone level, but this difference was not statistically significant. In addition, the HFP group presented a significantly increased testicular level of 17β-HSD, which is the key enzyme protein of testosterone biosynthesis (Figure 3).

### 3.4. LP1008 Improves Antioxidant Capacity and Suppresses Inflammation in the Testis

Antioxidant status and oxidative stress are assessed by measuring the levels of markers SOD, GPx, CAT, and MDA in the testes. As shown in Figure 4, the levels of SOD, GPx, and CAT in the HFC group were significantly lower than those in the NC group, and the lipid peroxide (MDA) concentration significantly increased. Moreover, LP1008 treatment significantly increased the GPx and CAT levels and decreased the MDA level; however, SOD activity tended to increase, but the difference was not statistically significant. The testicular inflammatory parameters are presented in Figure 5; the protein expression levels of TNF-α and NF-κB were significantly increased in the HFC group, and LP1008 treatment significantly decreased HFC-induced testicular inflammation, whereas no difference in IL-6 protein expression was observed. These results indicate that LP1008 alleviated HFD-induced oxidative stress and inflammation.

### 3.5. LP1008 Attenuates HFD-Induced Apoptosis and Autophagy

As shown in Figure 6 and Figure 7, the ratios of the proapoptotic factor Bax to the antiapoptotic factors Bcl-xl (Bax/Bcl-xl), cleaved caspase-3, PARP, cleaved PARP, and cleaved caspase-8 were significantly greater than those of the NC. Compared with the HFC group, treatment with LP1008 significantly ameliorated these apoptotic effects, with greater decreases in Bax, Bcl-xl, cleaved caspase-3, and PARP levels. Compared with those in the NC group, the LC3 and beclin-1 levels in the HFC group were significantly greater, and the P62 level was significantly lower, while LP1008 treatment attenuated the effects of autophagy, as indicated by the reversal of marker expression (Figure 8).

### 3.6. LP1008 Modulates the Gut Microbiota Composition of HFD-Fed Mice

Intestinal microorganisms are parasites that are affected by the intestinal structure. Multivariate analyses were implemented to compare the overall composition of the gut microbiota in terms of the OTU level. Alpha diversity analysis of microbial diversity and richness (the Shannon and Chao1 estimators) revealed that LP1008 significantly decreased the Shannon estimator, whereas there were no statistically significant differences in the Chao1 estimator among the three groups (Figure 9a,b). PCoA revealed a distinct clustering of the microbiota composition for each group (Figure 9c).

To further explore the effects of LP1008 on intestinal microbial species, the gut microbiota was analyzed using 16S rDNA pyrosequencing at the phylum and family levels. At the phylum level, *Firmicutes* and *Bacteroidetes* were the dominant phyla, and the HFC group presented a significantly greater ratio of *Firmicutes* to *Bacteroidetes*. LP1008 treatment also showed an increasing trend, but there was no significant difference when compared with the NC group (Figure 10a). At the family level, the abundances of *Lachnospiraceae* and *Ruminococcaceae* increased and that of *Muribaculaceae* decreased in the HFC group compared with those in the NC group, whereas LP1008 treatment decreased *Ruminococcaceae* and increased *Bifidobacteriaceae* (Figure 10b). Next, we applied the linear discriminant analysis (LDA) effect size (LEfSe) method to determine the bacteria that might explain the observed metabolic differences between the groups. The results revealed that the class *Clostridia*, order *Clostridiales*, family *Lachnospiraceae*, and species *Lachnospiraceae* bacterium 609 and genus *Lactococcus*, genus *Ruminiclostridium*, species *Ruminiclostridium* 9, and species *Ruminiclostridium* 5 were significantly more abundant in the HFC group than in the NC group, whereas the LP1008 treatment group presented significantly greater increases in the genera *Faecalibaculum*, *Erysipelatoclostridium*, genus *Blautia*, species *Romboutsia*, and species *Lachnospiraceae* FCS020 (Figure 10c).

## 4. Discussion

In recent decades, the main nutritional model of development has become the so-called Western diet, which includes high intakes of animal proteins, saturated and trans-fatty acids, and simple carbohydrates. An unhealthy diet may be directly associated with increased oxidative stress, being the underlying cause of obesity, type 2 diabetes, insulin resistance, and intestinal dysbiosis. Moreover, the above metabolic disorders have a negative impact on sperm quality [15]. In the present study, we observed that a HFD caused abnormal metabolic parameters, such as hypercholesterolemia, hyperglycemia, insulin resistance, and ALT and AST levels, in addition to hepatic steatosis, but the TG level was significantly decreased, which is consistent with previous findings [16]. LP1008 treatment ameliorated the HFD-induced increases in cholesterol, insulin resistance, and ALT and AST levels. Yoshitake et al. [11] reported that a HFD (62% fat) with heat-killed *L. plantarum* L-137 (HK L-137) supplementation resulted in decreased cholesterol and biomarkers of hepatic inflammation, AST and ALT, and tended to reduce plasma insulin.

A HFD may alter spermatogenesis and reproductive function, as shown in findings obtained in other studies [1,5,10]. The testicular testosterone concentration plays an important role in spermatogenesis. In this study, we found that the testosterone level and sperm quality, including sperm motility, sperm count, and sperm morphology, decreased more in the HFC group than in the control group, and morphological analysis revealed that the seminiferous epithelia were slightly thinned and disorganized. We also observed that the involvement of steroidogenesis enzymes such as 17β-HSD decreased. Our results were consistent with those of other studies, showing that a decreased expression of 17β-HSD is associated with decreased testosterone levels in HFD-fed rats [17]. We demonstrated that LP1008 treatment ameliorated the HFD-induced decreases in sperm quality and spermatogenesis.

Male infertility is strongly associated with reactive oxygen species (ROS), which are necessary for the entire process of spermatogenesis. However, high levels of ROS neutralize antioxidants in seminal plasma and cause oxidative stress [18]. LP1008 is an *L. plantarum* probiotic that has benefits in terms of mediating responses to oxidative stress, antioxidant activity, and inflammation [19]. In the present study, the levels of pro-inflammatory cytokines, such as LPS, NF-κB, and TNF-α, were increased, and the down-regulation of SOD, CAT, and GPx, which are antioxidative enzymes, was observed in the HFC group, indicating that the HFD-fed mice exhibited low-grade inflammation. Consistent with previous reports [20,21], LP1008 treatment ameliorated the imbalance in the antioxidant system and decreased inflammation in HFD-fed mice.

Spermatogenesis is a complex process that relies on coordinated cell proliferation and apoptosis, but excess apoptosis has been indicated to be defective in the process of spermatogenesis [1]. Feeding with a HFD could induce apoptosis, and LP1008 treatment reduced apoptosis. Autophagy is a catabolic pathway that transports nonessential, old, or damaged components to lysosomes for lysosome-mediated degradation and turnover [22]. We found that autophagy was increased in the HFC group, and that this increase was attenuated by LP1008 treatment. Huo et al. [5] reported active autophagy in HFD-fed mice and the impairment of the reproductive system, which suggested that the up-regulation of autophagy is associated with infertility. Autophagy and apoptosis often occur in the same cell, mostly in a sequence in which autophagy precedes apoptosis. Autophagy inhibits ER stress-induced cell apoptosis [23].

Studies have shown that the gut microbiota is an important environmental factor that contributes to the development of obesity, insulin resistance, and inflammation [24]. First, the present results indicated that the α diversity of the Shannon diversity of the gut microbiota was significantly decreased in the HFP group, and β diversity had three clusters in the PCoA plot. Low gut microbiota diversity is usually a hallmark of intestinal dysbiosis. Certain probiotics may increase competition for nutrients or alternatively lead to the production of antimicrobial peptides that reduce microbial growth [25]. At the phylum level, we found that the abundance of the intestinal microbiota *Firmicutes* increased, whereas the abundance of *Bacteroidetes* decreased, and the *Firmicutes*/*Bacteroidetes* ratio increased, which has been shown to be associated with changes in the gut microbiota in obese human and animal models [26]. However, LP1008 treatment led to an inverse trend without significant difference. Therefore, we further detected changes in gut microorganisms. At the family level, increases in *Lachnospiraceae* and *Ruminococcaceae* and a decrease in *Muribaculaceae* were observed in the HFC group. High abundances of *Lachnospiraceae* are positively correlated with glucose or lipid metabolism, indicating metabolic disturbance [27]. Kim et al. [28] also reported an increased abundance of *Ruminococcaceae* in HFD-fed mice compared with low-fat-diet (LFD)-fed mice. *Muribaculaceae* are also referred to as the family S24-7 and produce enzymes that degrade complex carbohydrates. Previous studies have demonstrated that *Muribaculaceae* can degrade carbohydrates, so high-calorie diets decrease the abundance of these bacteria [29].

LP1008 treatment significantly decreased *Ruminococcaceae* and improved *Bifidobacteriaceae,* which has been shown to reduce the LPS level and improve mucosal barrier functions [30]. In addition, biomarker analysis via linear discriminant analysis (LDA) effect size (LEfSe) in our study revealed that LP1008 treatment increased the abundance of the genera *Faecalibaculum*, *Blautia*, and *Lachnospiraceae* FCS020. *Faecalibaculum* is a widely known anti-inflammatory bacteria [31], and the results were consistent with those of Tang et al. [32], who reported an increasing abundance of *Faecalibaculum* with *L. plantarum* supplementation. *Blautia* has been demonstrated to be the most abundant bacteria in sperm and is also present in semen, supporting a possible contribution of upper genital tract microbes to the downstream seminal microbiome composition [33]. *Lachnospiraceae FCS020*, which are known butyrate producers, are correlated with fiber intake [34]

## 5. Conclusions

In summary, the *L. plantarum* strain used in this study comprehensively ameliorated metabolic disorders, including insulin resistance and dyslipidemia. *L. plantarum* improved metabolic disorders and reproductive function by reversing HFD-induced gut microbiota dysbiosis, with increased levels of antioxidative enzymes leading to the disruption of inflammation, autophagy, and apoptosis.

## Figures and Tables

**Figure 1 biology-13-00890-f001:**
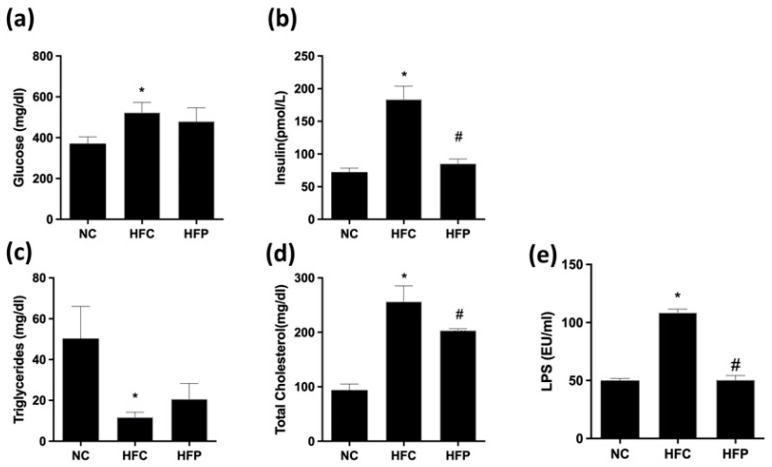
The effects of a high-fat diet and LP1008 treatment on the following serum biochemical parameters: (**a**) glucose, (**b**) insulin, (**c**) triglycerides, (**d**) total cholesterol, and (**e**) LPS in male mice. The data are expressed as the means ± SDs. * *p* < 0.05 compared with the NC group; ^#^ *p* < 0.05 compared with the HFC group. NC: normal control; HFC: high-fat control; HFP: high-fat diet + LP1008.

**Figure 2 biology-13-00890-f002:**
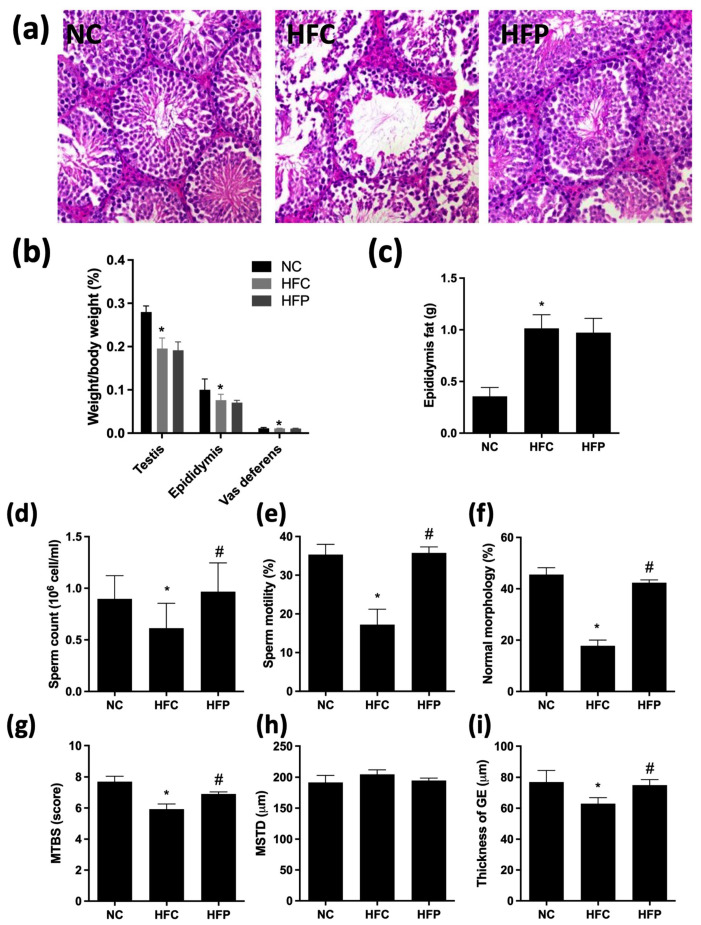
The effects of a high-fat diet and LP1008 treatment on (**a**) testicular histology; (**b**) relative weights of the testes, epididymis, and vas deferens; (**c**) epididymal fat; (**d**) sperm count; (**e**) sperm motility; (**f**) percentage of sperm with a normal morphology; (**g**) mean testicular biopsy score (MTBS); (**h**) mean seminiferous tubule diameter (MSTD); and (**i**) thickness of the germinal epithelium (GE) in male mice. Testicular sections were stained with hematoxylin and eosin. The data are expressed as the means ± SDs. * *p* < 0.05 compared with the NC group; ^#^ *p* < 0.05 compared with the HFC group. NC: normal control; HFC: high-fat control; HFP: high-fat diet + LP1008.

**Figure 3 biology-13-00890-f003:**
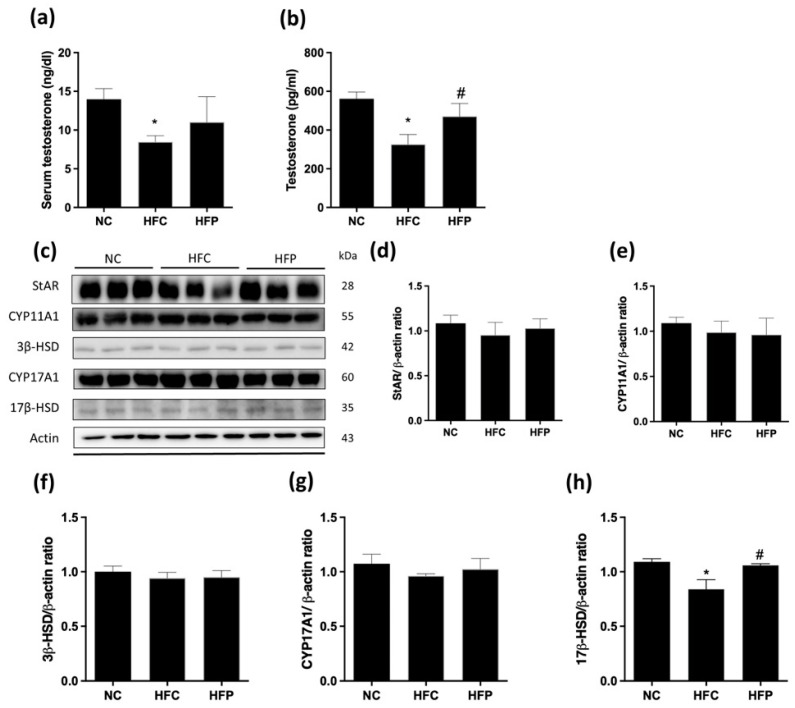
The effects of a high-fat diet and LP1008 treatment on the following testosterone biosynthesis regulators in male mice: (**a**) serum testosterone, (**b**) testicular testosterone, (**c**) protein expression in each group (Appendix A), (**d**) StAR, (**e**) CYP11A1, (**f**) 3β-HSD, (**g**) CYP17A1, and (**h**) 17β-HSD. Relative density analysis of protein bands in male mice. The data are expressed as the means ± SDs. * *p* < 0.05 compared with the NC group; ^#^ *p* < 0.05 compared with the HFC group. NC: normal control; HFC: high-fat control; HFP: high-fat diet + LP1008.

**Figure 4 biology-13-00890-f004:**
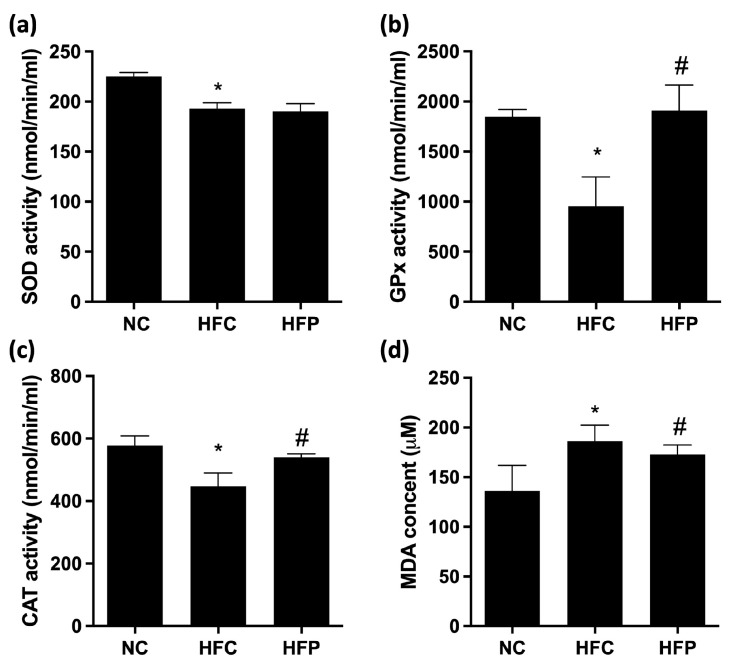
The effects of a high-fat diet and LP1008 treatment on (**a**) SOD activity, (**b**) GPx activity, (**c**) CAT activity, and (**d**) MDA content in male mice. The data are expressed as the means ± SDs. * *p* < 0.05 compared with the NC group; ^#^ *p* < 0.05 compared with the HFC group. NC: normal control; HFC: high-fat control; HFP: high-fat diet + LP1008.

**Figure 5 biology-13-00890-f005:**
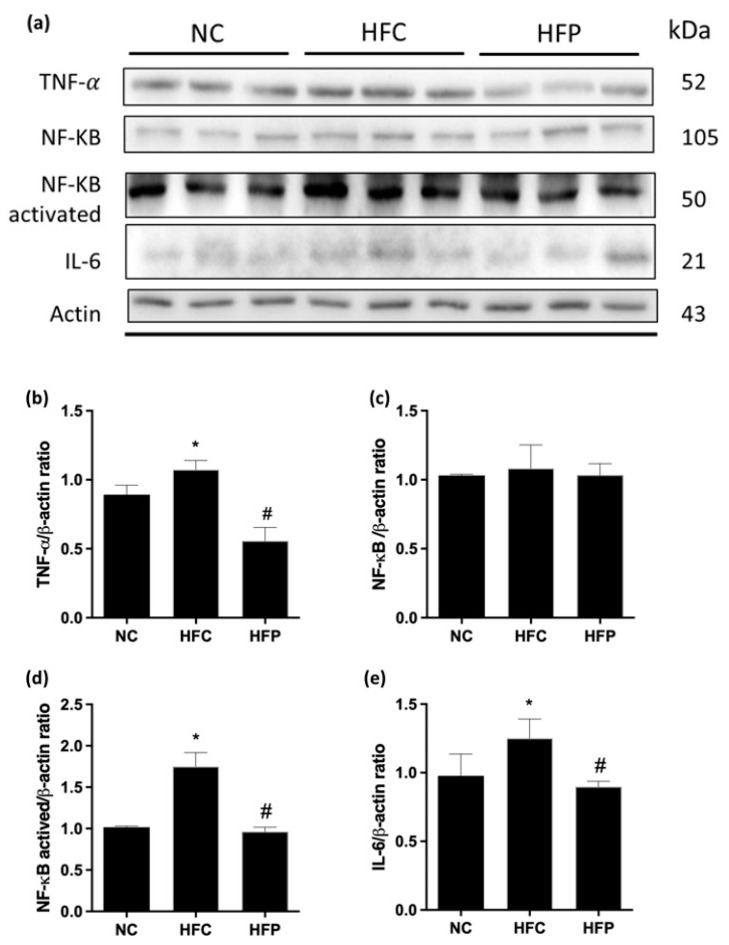
The effects of a high-fat diet and LP1008 treatment on (**a**) protein expression in each group (Appendix A) and (**b**) TNF-α, (**c**) NF-κB, (**d**) activated NF-κB, and (**e**) IL-6 levels in male mice. The data are expressed as the means ± SDs. * *p* < 0.05 compared with the NC group; ^#^ *p* < 0.05 compared with the HFC group. NC: normal control; HFC: high-fat control; HFP: high-fat diet + LP1008.

**Figure 6 biology-13-00890-f006:**
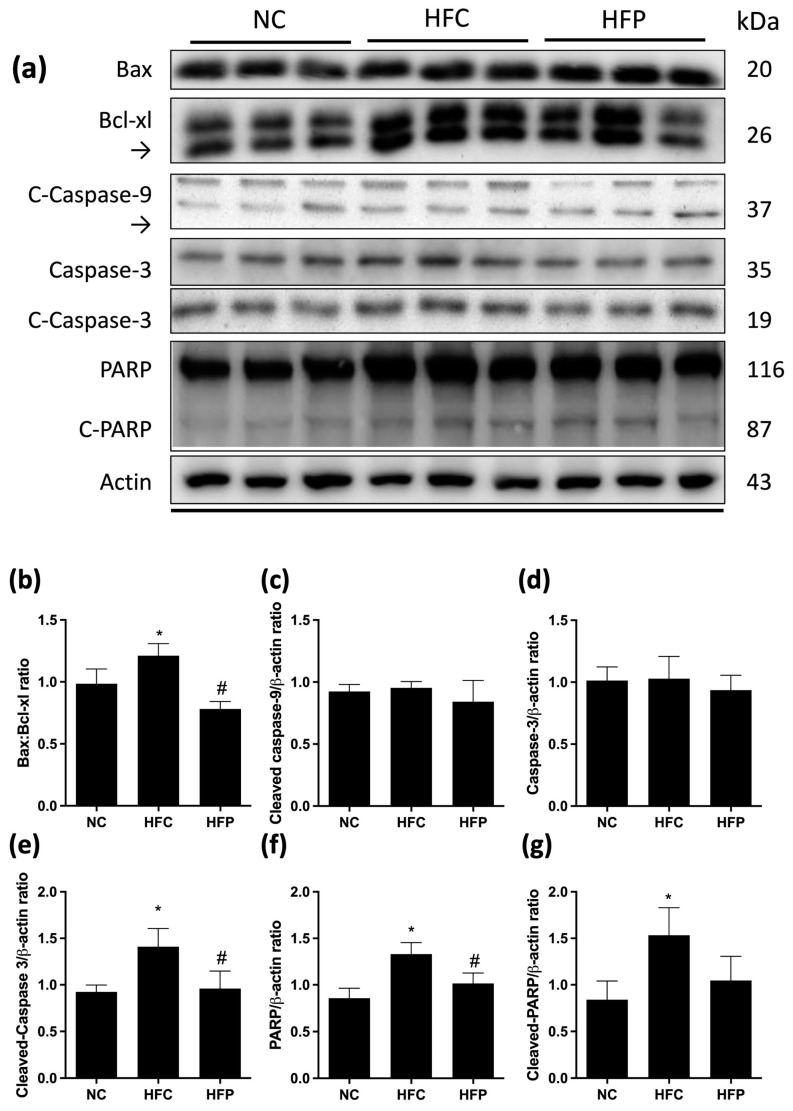
The effects of a high-fat diet and LP1008 treatment on testicular intrinsic apoptosis pathway regulators: (**a**) protein expression in each group (Appendix A), (**b**) Bax/Bcl-xl ratio, (**c**) cleaved caspase-9, (**d**) caspase-3, (**e**) cleaved caspase-3, (**f**) PARP, and (**g**) cleaved PARP in male mice. The data are expressed as the means ± SDs. * *p* < 0.05 compared with the NC group; ^#^ *p* < 0.05 compared with the HFC group. NC: normal control; HFC: high-fat control; HFP: high-fat diet + LP1008.

**Figure 7 biology-13-00890-f007:**
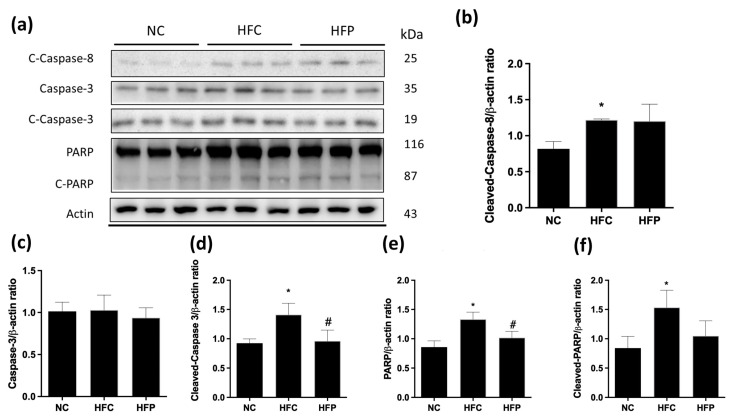
The effects of a high-fat diet and LP1008 treatment on testicular extrinsic apoptosis pathway regulators: (**a**) protein expression in each group (Appendix A), (**b**) cleaved caspase-8, (**c**) caspase-3, (**d**) cleaved caspase-3, (**e**) PARP, and (**f**) cleaved PARP in male mice. The data are expressed as the means ± SDs. * *p* < 0.05 compared with the NC group; ^#^ *p* < 0.05 compared with the HFC group. NC: normal control; HFC: high-fat control; HFP: high-fat diet + LP1008.

**Figure 8 biology-13-00890-f008:**
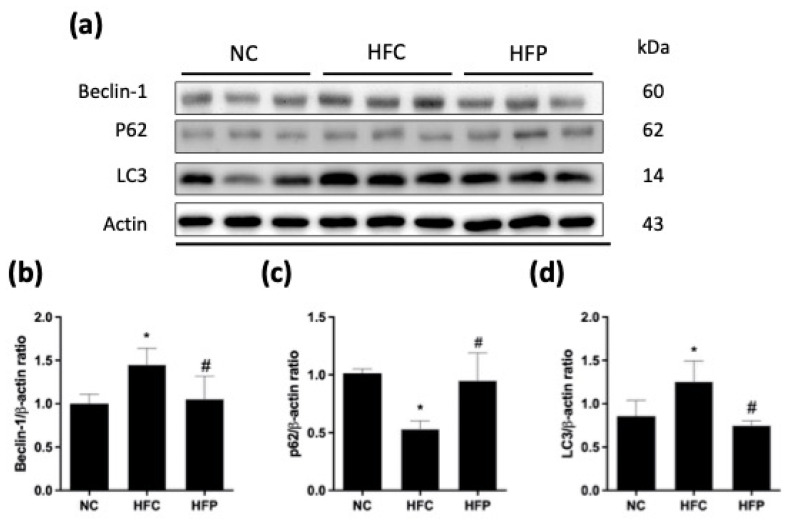
The effects of a high-fat diet and LP1008 treatment on testicular autophagy pathway regulators: (**a**) protein expression in each group (Appendix A), (**b**) beclin-1, (**c**) p62, and (**d**) LC3 in male mice. The data are expressed as the means ± SDs. * *p* < 0.05 compared with the NC group; ^#^ *p* < 0.05 compared with the HFC group. NC: normal control; HFC: high-fat control; HFP: high-fat diet + LP1008.

**Figure 9 biology-13-00890-f009:**
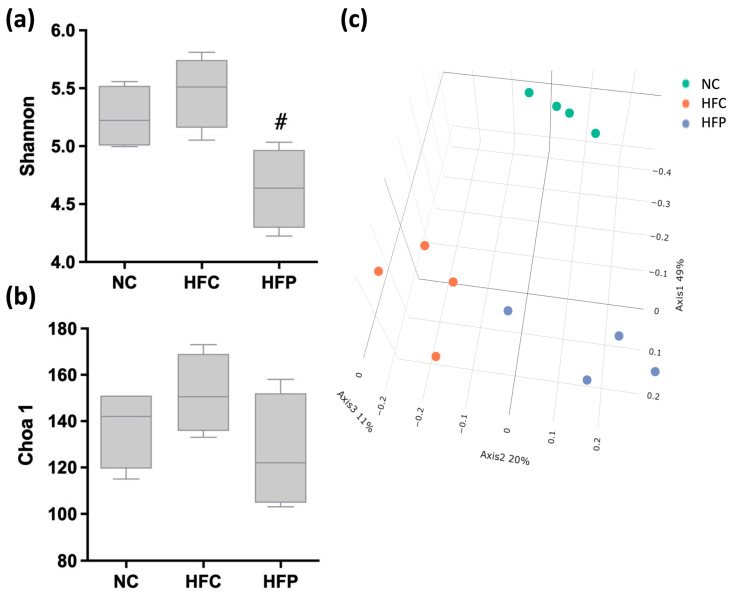
The effects of a high-fat diet and LP1008 treatment on the gut microbiota: (**a**,**b**) community diversity and richness assessed by the Shannon and Chao indices, (**c**) principal coordinate analysis (PCoA) score plot in male mice. ^#^ *p* < 0.05 compared with the HFC group. NC: normal control; HFC: high-fat control; HFP: high-fat diet + LP1008.

**Figure 10 biology-13-00890-f010:**
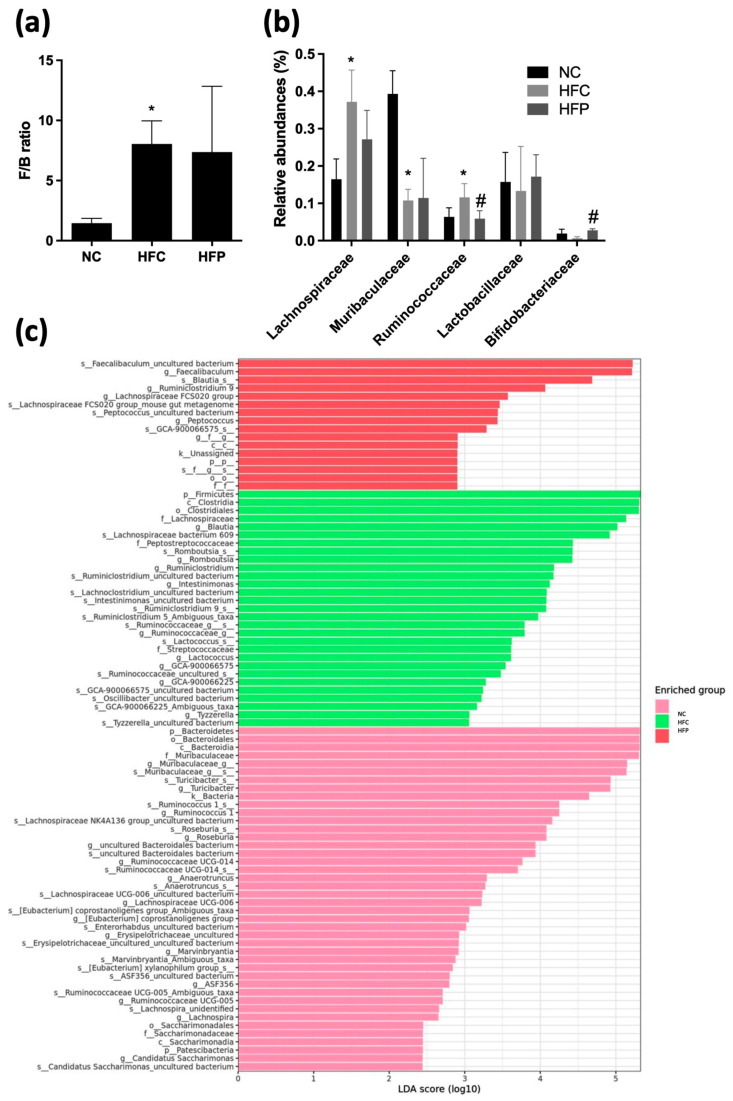
The effects of a high-fat diet and LP1008 treatment on the gut composition: (**a**) *Firmicutes/Bacteroidetes* (F/B) ratio, (**b**) abundances of the gut microbiota at the family level, and (**c**) linear discriminant analysis (LDA > 2) scores derived from LEfSe analysis in male mice. * *p* < 0.05 compared with the NC group; ^#^ *p* < 0.05 compared with the HFC group. NC: normal control; HFC: high-fat control; HFP: high-fat diet + LP1008.

## Data Availability

The original contributions presented in the study are included in the article/Appendix A, further inquiries can be directed to the corresponding authors.

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
