# Peer review of "Lactiplantibacillus plantarum 1008 Enhances Testicular Function and Spermatogenesis via the Modulation of Gut Microbiota in Male Mice with High-Fat-Diet-Induced Obesity"

_biology, 2024, doi:10.3390/biology13110890_

Round 1
Reviewer 1 Report
Comments and Suggestions for Authors
1. Brief Summary
The article aims to evaluate the effects of Lactobacillus plantarum 1008 (LP1008) on metabolic function, testicular antioxidant capacity, apoptosis, autophagy, and spermatogenesis in male mice with diet-induced obesity. The main contributions include demonstrating that the mouse model with diet-induced obesity was capable of causing alterations in serum biochemical parameters, low testicular testosterone levels, poor sperm quality, and 17β-HSD protein expression, as well as increased apoptosis, inflammation, and autophagy proteins in the testis. A significant strength of the article is its demonstration that the use of the probiotic LP1008 increased antioxidant enzyme levels, regulated the microbiota, improved high-fat diet (HFD)-induced oxidative stress, apoptosis, inflammation, and autophagy, thereby enhancing testicular function and semen quality.
2. General Concept Comments
This reviewer commends the authors for an excellent article. Precision medicine has advanced significantly with the use of customized laboratory animals. This article represents an important step in developing effective treatments to combat male infertility and address obesity-related issues. In addition to the specific points listed below, this reviewer highlights the necessity of including citations in the text regarding the procedures performed on laboratory animals, emphasizing compliance with animal welfare practices authorized in the country.
3. Specific Comments
1 - It would be important to include the initial and final weight of the mice in each treatment group to enable further analyses, such as the gonadosomatic index.
2 - It is important to mention that the anesthetic protocol used is the one indicated for euthanasia.
3 - It is necessary to improve the resolution of Figure 6.
Author Response
Dear Professor Editor:
Please find enclosed our revised original paper entitled “Lactiplantibacillus plantarum 1008 Enhances Testicular Function and Spermatogenesis via Modulating Gut Microbiota in Male Mice with High-Fat Diet-Induced Obesity”. We appreciated the comments and suggestions provided to further improve our manuscript.
Sincerely yours
Chih-Wei Tsao, MD., Ph.D.
Division of Urology, Department of Surgery, Tri-Service General Hospital, National Defense Medical Center
No. 325, Section 2, Cheng-Gung Road,
Neihu, Taipei 114, Taiwan
Telephone: +886-2-87927170
Fax: +886-2-87927172
e-mail: weisurger@gmail.com
Reviewer 1
Comments and Suggestions for Authors
- Brief Summary
The article aims to evaluate the effects of Lactobacillus plantarum 1008 (LP1008) on metabolic function, testicular antioxidant capacity, apoptosis, autophagy, and spermatogenesis in male mice with diet-induced obesity. The main contributions include demonstrating that the mouse model with diet-induced obesity was capable of causing alterations in serum biochemical parameters, low testicular testosterone levels, poor sperm quality, and 17β-HSD protein expression, as well as increased apoptosis, inflammation, and autophagy proteins in the testis. A significant strength of the article is its demonstration that the use of the probiotic LP1008 increased antioxidant enzyme levels, regulated the microbiota, improved high-fat diet (HFD)-induced oxidative stress, apoptosis, inflammation, and autophagy, thereby enhancing testicular function and semen quality.
- General Concept Comments
This reviewer commends the authors for an excellent article. Precision medicine has advanced significantly with the use of customized laboratory animals. This article represents an important step in developing effective treatments to combat male infertility and address obesity-related issues. In addition to the specific points listed below, this reviewer highlights the necessity of including citations in the text regarding the procedures performed on laboratory animals, emphasizing compliance with animal welfare practices authorized in the country.
- Specific Comments
1 - It would be important to include the initial and final weight of the mice in each treatment group to enable further analyses, such as the gonadosomatic index.
Our response:
Thank you for professional comment. The results of initial and final weight of the mice in each treatment group were added in the statement of “Results” 3.2. Moreover we revised the original Figure 2 by adding the gonadosomatic index.
2 - It is important to mention that the anesthetic protocol used is the one indicated for euthanasia.
Our response:
Thank you for your kind suggestion. The protocol has been revised in lines 92-101. After 8 weeks of gavage the mice were deeply anesthetized with a mixture of Zoletil 50 (containing 25 mg/mL tiletamine and zolazepam; Virbac, Carros, France) and Rompun (containing 23.32 mg/mL xylazine hydrochloride; Bayer, Taipei, Taiwan) administered intraperitoneally. Upon confirmation of adequate anesthesia through the absence of reflexes, exsanguination was conducted via cardiac puncture. A sterile 25-gauge needle attached to a 1 mL syringe was carefully inserted into the left ventricle, and blood was withdrawn slowly to prevent hemolysis. The collected blood was immediately processed for downstream analyses. This procedure ensured minimal stress and discomfort to the animals and was performed as part of humane euthanasia at the end of the experimental protocol.
3 - It is necessary to improve the resolution of Figure 6.
Our response:
Thank you for your valuable suggestion. We have improved the resolution of Figure 6 as recommended. Please find the updated figure in the revised manuscript.
Submission Date
25 September 2024
Date of this review
15 Oct 2024 19:34:32

Reviewer 2 Report
Comments and Suggestions for Authors
The present study illustrates the impact of Lactobacillus plantarum on gut microbiota and its subsequent effect on reproductive reversal in obesity-induced mice. The study is well-designed, highlighting the carry-over effect of Lactobacillus plantarum against an obesity-induced model. The inclusion of such remedies is timely, as they can improve bodily functions without altering gut microbiota. The data included is sufficient and has the potential for publication; however, revisions are needed to enhance the content.
Title: Modify the title to emphasize the role of gut microbiota, as it is a key element in this study. Reconsider using the word "retrieve" in the title.
Introduction: Include sentences explaining how obesity interacts with male reproductive disorders. Also, discuss which types of remedies have been tested for reproductive performance in obesity models and their possible drawbacks.
Materials and Methods:
- Revise subheading 2.1. to "Animal Management and Experimental Groups."
- For subheading 2.3., clarify how sperm samples were collected.
- Explain the procedure for gut microbiota sampling.
- Provide details on the number of original sequences obtained for each group and the number of effective reads after filtering out unqualified sequences.
Results and Discussion: Adequately presented.
Conclusions: Should be based on the major findings of the study rather than a general statement.
Author Response
Dear Professor Editor:
Please find enclosed our revised original paper entitled “Lactiplantibacillus plantarum 1008 Enhances Testicular Function and Spermatogenesis via Modulating Gut Microbiota in Male Mice with High-Fat Diet-Induced Obesity”. We appreciated the comments and suggestions provided to further improve our manuscript.
Sincerely yours
Chih-Wei Tsao, MD., Ph.D.
Division of Urology, Department of Surgery, Tri-Service General Hospital, National Defense Medical Center
No. 325, Section 2, Cheng-Gung Road,
Neihu, Taipei 114, Taiwan
Telephone: +886-2-87927170
Fax: +886-2-87927172
e-mail: weisurger@gmail.com
Reviewer2
Comments and Suggestions for Authors
The present study illustrates the impact of Lactobacillus plantarum on gut microbiota and its subsequent effect on reproductive reversal in obesity-induced mice. The study is well-designed, highlighting the carry-over effect of Lactobacillus plantarum against an obesity-induced model. The inclusion of such remedies is timely, as they can improve bodily functions without altering gut microbiota. The data included is sufficient and has the potential for publication; however, revisions are needed to enhance the content.
Title: Modify the title to emphasize the role of gut microbiota, as it is a key element in this study. Reconsider using the word "retrieve" in the title.
Our response:
Thank you for your valuable feedback regarding the title. In response, we have revised it to better emphasize the role of gut microbiota as a key element in the study. The updated title is: " Lactiplantibacillus plantarum 1008 Enhances Testicular Function and Spermatogenesis via Modulating Gut Microbiota in Male Mice with High-Fat Diet-Induced Obesity"
Introduction: Include sentences explaining how obesity interacts with male reproductive disorders. Also, discuss which types of remedies have been tested for reproductive performance in obesity models and their possible drawbacks.
Our response:
Thank you for the academic suggestions. The revisions to the introduction are marked in lines 52-56 and 71-80.
Materials and Methods:
- Revise subheading 2.1. to "Animal Management and Experimental Groups."
Our response:
Thank you for the reviewer’s suggestions. The revisions are marked in line 83.
- For subheading 2.3., clarify how sperm samples were collected.
Our response:
Thank you for the reviewer’s suggestions. The revisions are marked in lines 119-120.
- Explain the procedure for gut microbiota sampling.
Ourresponse:
Thank you for the reviewer’s suggestions. The revisions are marked in lines 163-169.
- Provide details on the number of original sequences obtained for each group and the number of effective reads after filtering out unqualified sequences.
Our response:
Thank you for the reviewer’s suggestions. The revisions are marked in lines 174-176.
Results and Discussion: Adequately presented.
Conclusions: Should be based on the major findings of the study rather than a general statement.
Our response:
Thank you for the reviewer’s suggestions. We have revised the statement of “Conclusions” part.
Submission Date
25 September 2024
Date of this review
06 Oct 2024 14:09:07
